# Automating modeling in mechanics: LLMs as designers of physics-constrained neural networks for constitutive modeling of materials

## Abstract

Large language model (LLM)-based agentic frameworks increasingly adopt the paradigm of dynamically generating task-specific agents. We suggest that not only agents but also specialized software modules for scientific and engineering tasks can be generated on demand. We demonstrate this concept in the field of solid mechanics. There, so-called constitutive models are required to describe the relationship between mechanical stress and body deformation. Constitutive models are essential for both the scientific understanding and industrial application of materials. However, even recent data-driven methods of constitutive modeling, such as constitutive artificial neural networks (CANNs), still require substantial expert knowledge and human labor. We present a framework in which an LLM generates a CANN on demand, tailored to a given material class and dataset provided by the user. The framework covers LLM-based architecture selection, integration of physical constraints, and complete code generation. Evaluation on three benchmark problems demonstrates that LLM-generated CANNs achieve accuracy comparable to or greater than manually engineered counterparts, while also exhibiting reliable generalization to unseen loading scenarios and extrapolation to large deformations. These findings indicate that LLM-based generation of physics-constrained neural networks can substantially reduce the expertise required for constitutive modeling and represent a step toward practical end-to-end automation.

## 1 Introduction

Automation in constitutive modeling:

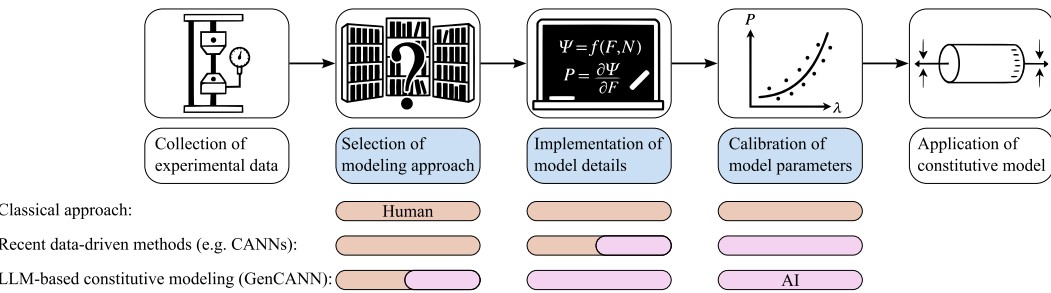

Figure 1: Evolution of constitutive modeling in solid mechanics over time. Until the 2010s, scientists manually derived constitutive models to describe experimental observations. Recent data-driven methods, such as constitutive artificial neural networks (CANNs), partially automate model implementation and fully automate model calibration. Our framework leverages LLMs to generate CANNs on demand, pushing towards complete end-to-end automation.

Constitutive models capture our understanding of how materials behave under mechanical load, expressed as mathematical relationships linking stresses to strains. Calibrated with experiments, they predict behavior beyond what was directly tested, including complex cases that are hard or impossible to reproduce in the lab. These predictions allow for realistic mechanical simulations of engineered products and biological tissues, which deepen scientific understanding and reduce the time and cost of component design.

Historically, constitutive behavior was captured by empirically derived symbolic laws such as Mooney–Rivlin, Neo Hookean, and Ogden (Mooney, 1940; Rivlin, 1948a;b; Ogden, 1972). To reduce the effort of handcrafted laws, data-driven approaches arose: distance minimizing data-driven computing (Kirchdoerfer & Ortiz, 2016; Carrara et al., 2020), black-box surrogates via neural networks (Ghaboussi et al., 1991; Hashash et al., 2004), and spline-based interpolants (Sussman & Bathe, 2009; Latorre & Montáns, 2013; Dal et al., 2023). These are typically flexible but data hungry, weak at extrapolation, and difficult to interpret.

Gray box strategies embed physics to improve reliability (Fuhg et al., 2024): PINNs (Hao et al., 2022); MIANNs/PANNs that hard enforce mechanics (As' ad et al., 2022; Linden et al., 2023); and, central to our benchmarks, CANN families that blend constitutive structure with learning (Linka et al., 2021; 2023a; McCulloch et al., 2024; Abdolazizi et al., 2024). Related hybrids add neural corrections to mechanistic baselines (FuCe (Tushar et al., 2025)) or learn path dependence via neural ODEs (Taç et al., 2023). These approaches cut data needs and aid extrapolation, yet retain black box elements.

In parallel, interpretable methods seek explicit, inspectable laws: symbolic regression (Koza, 1993; Abdusalamov et al., 2023); EUCLID style inference from fields and forces (Flaschel et al., 2021; 2022; Joshi et al., 2022; Thakolkaran et al., 2022); and KAN based models that yield closed-form constitutive expressions, including CKANs (Kolmogorov, 1961; Liu et al., 2024c;b; Abdolazizi et al., 2025). Despite growing automation, effective use still demands substantial expertise.

LLM-based code generation lowers this barrier by automatically assembling the data processing, model setup, and solver code needed to build simulation or optimization pipelines from plain-language task descriptions. Such frameworks appear across diverse domains, including engineering optimization, PDEs, graph and materials modeling, and chemical engineering (Rios et al., 2024; Hao et al., 2024; Wuwu et al., 2025; Li et al., 2025; Verma et al., 2025; Huang et al., 2025; Heyer et al., 2025). A recent thrust centers on bilevel optimization, where an LLM-driven outer loop proposes solution candidates and an inner loop performs numerical calibration and evaluation (Chen et al., 2024a; Pandey et al., 2025). Within this pattern, the scientific generative agent (SGA) applies this approach to general scientific hypothesis generation (Ma et al., 2024), while the constitutive scientific generative agent (CSGA) adapts it for constitutive modeling (Tacke et al., 2025). Across benchmark stress–strain prediction tasks, CSGA outperforms SGA but remains less accurate than highly specialized methods such as constitutive artificial neural networks (CANNs).

Beyond single-LLM pipelines, agentic systems use multiple LLMs to plan, write, execute, and refine domain-specific code, such as MechAgents for finite-element mechanics (Ni & Buehler, 2024) and MDAgent for molecular dynamics (Shi et al., 2025). These frameworks are evolving from fixed teams to dynamic, task-specific organizations through subtask decomposition and specialized subagents (Wang et al.; Chen et al., 2024b), and even toward self-developing 'agent OS' platforms (Tang et al., 2025). Related work explores adaptive teaming and coordination (Liu et al., 2024a; Nettem et al., 2025) or frames agent design as evolutionary search (Yuan et al., 2025).

Motivated by dynamic agent generation, we propose an LLM-driven framework that creates task-specific constitutive artificial neural networks (CANNs) on demand and then immediately uses these self-generated new modules. While the constitutive scientific generative agent (CSGA) improved usability by letting non-experts build constitutive models with LLMs, it lacked the accuracy of specialized CANNs. Our approach combines both strengths: the LLM automatically designs, configures, and calibrates a CANN tailored to each material, offering the simplicity of an LLM interface and the accuracy of CANNs. We call these LLM-generated networks GenCANNs and refer to human-designed ones simply as CANNs. The progression toward automation enabled by GenCANNs is illustrated in Figure 1. The remainder of the paper is organized as follows: Section 2 provides background and introduces benchmark methods. Section 3 describes our approach. Section 4 reports evaluation results, and Section 5 discusses findings and concludes the paper.

## 2 BACKGROUND

### 2.1 CONTINUUM MECHANICS ESSENTIALS

Because constitutive models are typically formulated in the context of continuum mechanics, we summarize the essentials for this work here. For a comprehensive overview, see Holzapfel (2001). Material points are labeled by their reference position $\mathbf{X}$ and current position $\mathbf{x}$. The deformation is characterized by the deformation gradient $\mathbf{F}$ and the right Cauchy–Green deformation tensor $\mathbf{C}$:

$$\mathbf{F} = \frac{\partial \mathbf{x}}{\partial \mathbf{X}}, \qquad \mathbf{C} = \mathbf{F}^T \mathbf{F}.$$

In simple loading cases, such as uniaxial tension, the deformation can be described by the stretch $\lambda = \frac{l}{l_0}$, which is the ratio of current length $l$ to reference length $l_0$ and, in this case, corresponds to the relevant diagonal entry of $\mathbf{F}$. For simple shear, where material layers undergo a lateral displacement, the deformation is often characterized by the shear $\gamma = \frac{u}{h}$, the ratio of lateral displacement $u$ to specimen height $h$, which then coincides with the corresponding off-diagonal entry of $\mathbf{F}$.

The scalar invariants of $\mathbf{C}$ are $I_1$, $I_2$, and $I_3$. Incompressibility means $\det(\mathbf{C}) = 1$, hence $I_3 = 1$:

$$I_1 = \text{tr}(\mathbf{C}), \qquad I_2 = \frac{1}{2}(\text{tr}(\mathbf{C})^2 - \text{tr}(\mathbf{C}^2)), \qquad I_3 = \det(\mathbf{C}) = 1.$$

For isotropic materials, the strain-energy density $\Psi$ depends only on invariants of $\mathbf{C}$, whereas anisotropy requires an additional description of the form of anisotropy. In this work, we only assume transverse isotropy with a single preferred fiber direction $\mathbf{n}$, define the structure tensor $\mathbf{N}$, and form the additional invariants $I_4$ and $I_5$:

$$\mathbf{N} = \mathbf{n} \otimes \mathbf{n}, \qquad I_4 = \mathbf{N} : \mathbf{C}, \qquad I_5 = \mathbf{N} : \mathbf{C}^2.$$

A material is considered hyperelastic if its mechanical behavior can be described by a strain energy density function, denoted as $\Psi$. In this work, we exclusively focus on the concept of hyperelastic materials, which describes rubber and various types of biological tissue in many situations with satisfactory accuracy. The task of constitutive modeling is to define the strain energy $\Psi$ as a function $f$ of the deformation state, that is, $\mathbf{F}$ or $\mathbf{C}$. Once the strain energy function $\Psi$ is known, the isochoric part of the first Piola–Kirchhoff stress, $\mathbf{P}_{iso}$, can be determined. The incompressibility constraint adds a volumetric term, $-p\mathbf{F}^{-T}$, to the total stress $\mathbf{P}$, where $p$ serves as a Lagrange multiplier enforcing incompressibility and corresponds to the hydrostatic pressure:

$$\Psi = f(I_1, I_2, I_4, I_5), \qquad \mathbf{P}_{iso} = \frac{\partial \Psi}{\partial \mathbf{F}}, \qquad \mathbf{P} = \frac{\partial \Psi}{\partial \mathbf{F}} - p\mathbf{F}^{-T},$$

The first Piola–Kirchhoff stress $\mathbf{P}$ represents the load per unit area in the undeformed reference configuration, whereas the Cauchy stress $\sigma$ refers to the load per unit area in the deformed spatial configuration. $\sigma$ can be computed from $\mathbf{P}$ using the deformation gradient $\mathbf{F}$ and its determinant $J$:

$$J = \det(\mathbf{F}), \qquad \sigma = J^{-1}\mathbf{P}\mathbf{F}^T.$$

Both $\mathbf{P}$ and $\sigma$ are second-order tensors, typically represented in 3D as 3×3 matrices, where $\mathbf{X}_{ij}$ denotes the entry in row $i$ and column $j$.

### 2.2 CONSTITUTIVE ARTIFICIAL NEURAL NETWORKS (CANNS)

As we aim to generate constitutive artificial neural networks (CANNs) on demand using large language models (LLMs), we compare them against CANNs designed by human experts. These models follow the continuum mechanics framework outlined in Section 2.1. They are gray-box models: wherever possible, they incorporate white-box relationships from continuum mechanics, while using a black-box feed-forward neural network to predict strain energy from invariants of the deformation and structure tensors. Stresses are then computed by differentiating this energy with respect to the deformation. This approach reduces the tensor-to-tensor mapping to a compact scalar regression, enforces thermodynamic consistency, and improves interpretability. For each dataset, we compare with the most accurate published CANN variants (Linka et al., 2021; Pierre et al., 2023; Linka et al., 2023b). Re-implementing or adapting CANNs remains challenging, as details such as input selection and constraint enforcement must be tailored for each material. Consequently, constitutive modeling in the current paradigm and prior to GenCANN still requires deep expertise.

### 2.3 CONSTITUTIVE SCIENTIFIC GENERATIVE AGENT (CSGA)

An obvious route to automate constitutive modeling is to use LLMs. Using an LLM as a direct strain-to-stress surrogate is unreliable and misses its strength in code generation. The scientific generative agent (SGA) addresses this by letting an LLM propose, implement, and refine constitutive models. The constitutive scientific generative agent (CSGA) specializes SGA for continuum mechanics by adding assumptions (e.g., isotropy, incompressibility), defining inputs and outputs, suggesting an invariant basis, and enforcing zero stress at the reference state. We use CSGA as our second benchmark, complementing CANNs, as it is the most specialized LLM-based framework for constitutive modeling so far. The agent runs the code, receives loss feedback, and revises the model. Reported studies (Tacke et al., 2025) show that CSGA outperforms SGA but remains less accurate than CANNs. Its advantage lies in ease of use via a plain-text interface.

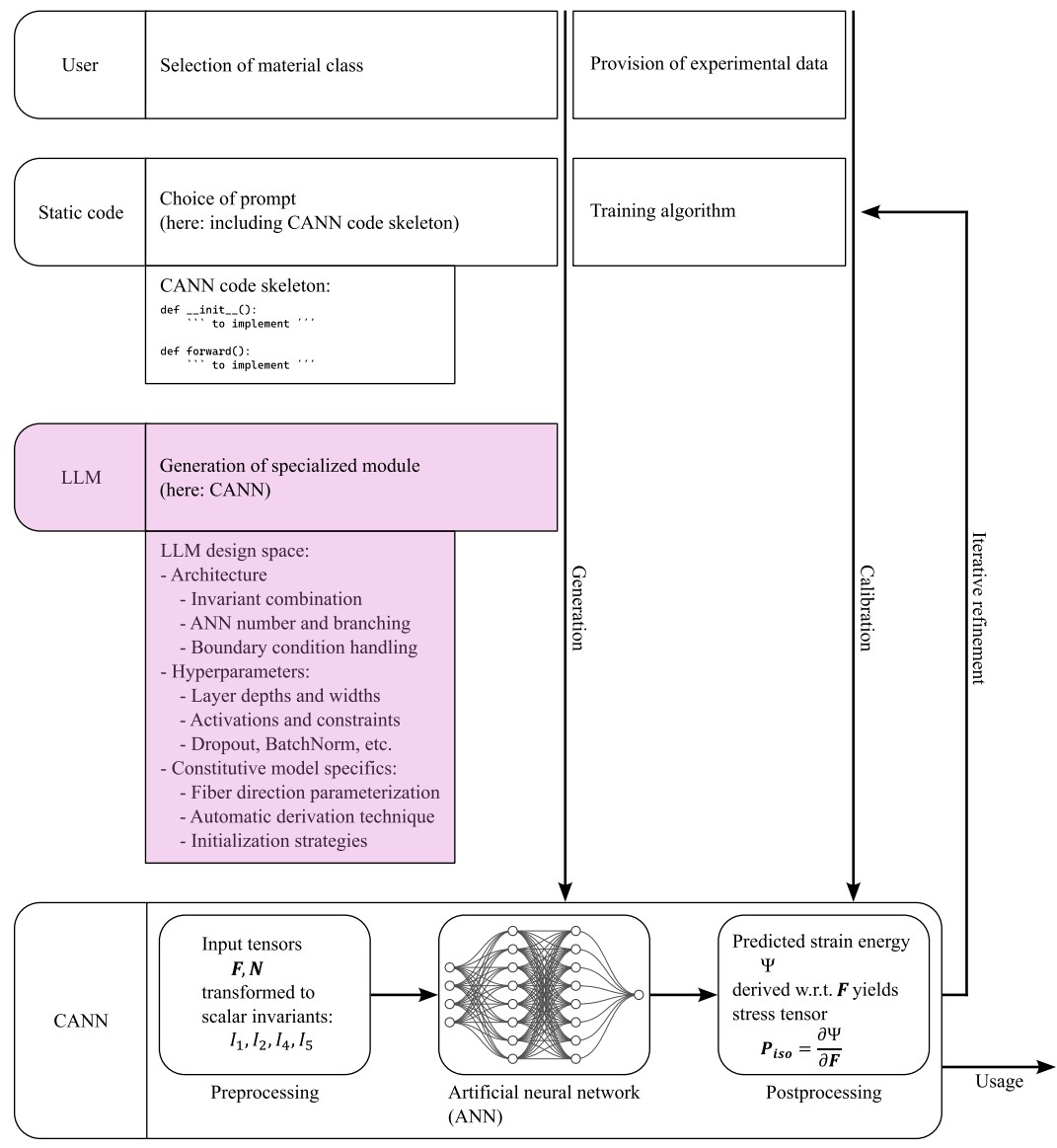

Figure 2: Process of the LLM-based generation of a constitutive artificial neural network (CANN). Static code translates the material classification into a prompt that includes a task description, continuum mechanics theory, formal requirements, and a rough CANN code skeleton, prompting the LLM to implement a CANN tailored to the specific modeling task. The resulting CANN is then trained and evaluated on experimental data and iteratively refined.

## 3 METHOD

Current approaches to constitutive modeling follow two paths. Specialized models such as CANNs are highly accurate and inherently satisfy physical constraints such as objectivity and thermodynamic consistency but are difficult to implement. In contrast, LLM-based agents like CSGA are easy to use through a text interface but lack accuracy and do not enforce these constraints. Rather than replacing one with the other, we combine their strengths: the LLM generates, on demand and from scratch, a CANN tailored to the material at hand. We refer to this generated model as GenCANN, short for LLM-generated CANN. This way, the LLM builds on (instead of competes with) decades of research, offering the simplicity of an LLM interface together with the accuracy and consistency of CANNs.

At the core of our pipeline is a large language model, OpenAI's O3, which we use without additional training. We focus on hyperelastic incompressible materials that are either isotropic or transversely isotropic. Figure 2 summarizes the LLM's role in the CANN design process. The LLM receives a two-part prompt. The first part describes the task: to implement a CANN that matches the chosen material class and follows certain coding requirements. It also includes a short summary of the continuum mechanics theory that links stress and strain through strain energy, similar to the Background Section 2.1. The second part is a compact code skeleton that guides the implementation. For isotropy, the skeleton defines the signatures of the classes CANN, PsiLayer, PartialPsiLayer, and the method build_cann_model(). For transverse isotropy, it also includes a StructureTensor stub. These small differences are intentional and practical because users can usually decide easily whether a material is isotropic or has a single preferred fiber direction. Turning that decision into a correct implementation is the hard part, and our approach automates it. The generated CANN combines three elements: preprocessing, one or more feedforward neural networks, and postprocessing. All of these are implemented by the LLM. Preprocessing and postprocessing, which include tensor assembly, invariant computation, and stress derivation, are mostly determined by continuum mechanics theory. The main design freedom lies in the feedforward neural networks that map invariants to strain energy. For these networks, the LLM decides on invariant combinations, network architecture and size, activation functions, constraints and regularization, handling and estimation of fiber directions when needed, weight initialization, and treatment of boundary conditions. Once the CANN is implemented, it is executed and trained on the provided data. The complete script and its $R^2$ score are sent back to the LLM for three refinement rounds. The best-performing version is kept as the final model. We repeat the complete CANN generation process five times per dataset, present the statistical analysis in Figure 9, and show the best-performing CANNs in Figures 3–8. An exemplary LLM-generated CANN implementation is shown in Section A.3.

## 4 RESULTS

### 4.1 BRAIN DATA

We begin with a dataset on the mechanical behavior of human brain tissue, an established benchmark for hyperelastic constitutive modeling (Budday et al., 2017a;b; 2019). Accurate models support impact simulation, injury prediction, and protective design. Brain tissue is soft, nearly incompressible, strain-stiffening, and asymmetric in tension and compression. The data were collected by Budday et al. (2017a) through mechanical tests on specimens excised from ten post-mortem human brains (7 male, 3 female, ages 54–81) within 60 hours of death. Multiple regions were sampled, we focus on their cortical gray matter. The tissue was subjected to three loading modes: uniaxial tension, uniaxial compression, and simple shear. For each loading mode, specimens underwent loading–unloading cycles, and the mean stress over the hysteresis loop was taken as the effective elastic response. 17 stress–strain points were reported for each loading mode.

We use the best CANN reported in the literature (Pierre et al., 2023), selected from multiple CANN optimization studies on this dataset (Budday et al., 2019; Linka et al., 2023a; Pierre et al., 2023; McCulloch et al., 2024), and the previously introduced CSGA (Tacke et al., 2025) as benchmarks. Across uniaxial tension, compression, and simple shear, all three methods closely reproduce the measured stress–strain response, as shown in Figure 3. Table 1 summarizes $R^2$ scores across all datasets. All models achieve $R^2$ scores above 0.90 on the brain tissue dataset, with only CSGA showing one score below 0.95. All three approaches predict training and test points reliably, but

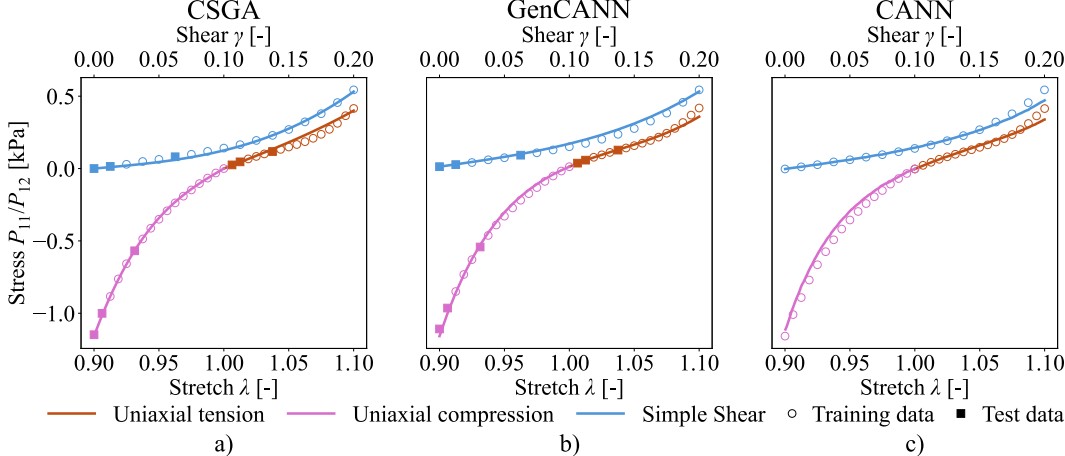

Figure 3: Predictions of the LLM-generated constitutive artificial neural network (GenCANN) for mechanical stress induced by brain tissue deformation compared against two benchmarks: the LLM-based constitutive scientific generative agent (CSGA) and the human-designed CANN.

were trained on all tested loading conditions. These results confirm that each model captures the complex behavior in the training data, but do not show generalization to unseen loading conditions.

## 4.2 RUBBER DATA

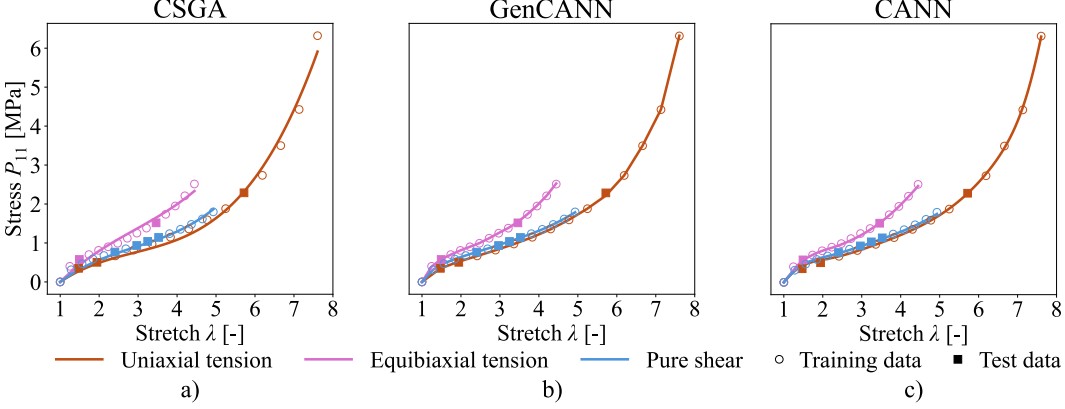

Figure 4: Predictions of the LLM-generated constitutive artificial neural network (GenCANN) for mechanical stress induced by rubber deformation compared against two benchmarks: the LLM-based constitutive scientific generative agent (CSGA) and the human-designed CANN.

Rubber is the classic example of a hyperelastic solid, capable of large, reversible strains beyond the scope of linear elasticity. Accurate modeling enables reliable design of components like tires and seals. We study two datasets: Treloar's classic experiments (Treloar, 1944) and a separate synthetic dataset that represents a similar fictitious material, provides ground truth for complex loading scenarios, and was introduced in the first publication on CANNs (Linka et al., 2021). Both cover uniaxial tension, equibiaxial tension, and pure shear, with 15 samples per protocol, and we keep the train–test split used in the first CANN publication. The experimental data anchor the problem in reality but span only a few loading paths, so a central question is how well models extrapolate to mixed multiaxial states not seen during training. Measuring such states in experiments is often not possible. The synthetic dataset addresses this by computing exact stresses for arbitrary deformations from an isotropic, incompressible rubber-like material, enabling a clean assessment of generalization beyond the trained loading paths. For both rubber datasets, we use the optimal CANN from its initial publication (Linka et al., 2021) and the CSGA (Tacke et al., 2025) as benchmarks.

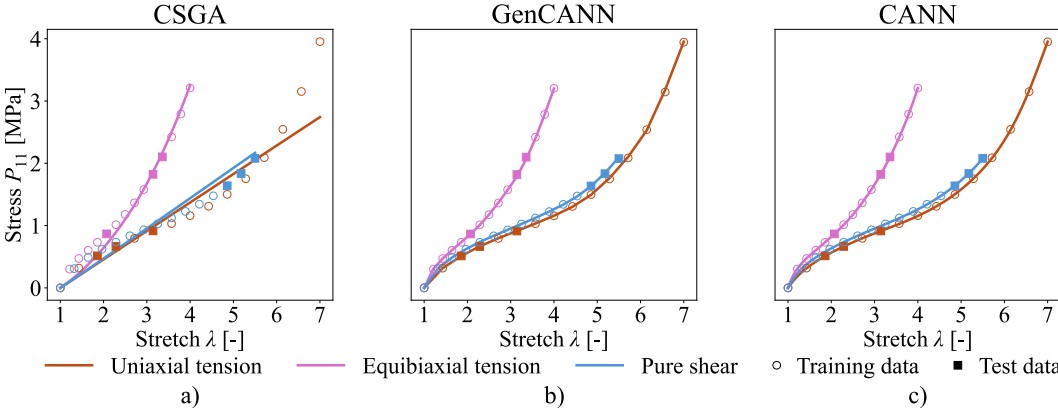

Figure 5: Predictions of the LLM-generated constitutive artificial neural network (GenCANN) for mechanical stress induced by deformation of a fictitious rubber-like material compared against two benchmarks: the LLM-based constitutive scientific generative agent (CSGA) and the human-designed CANN.

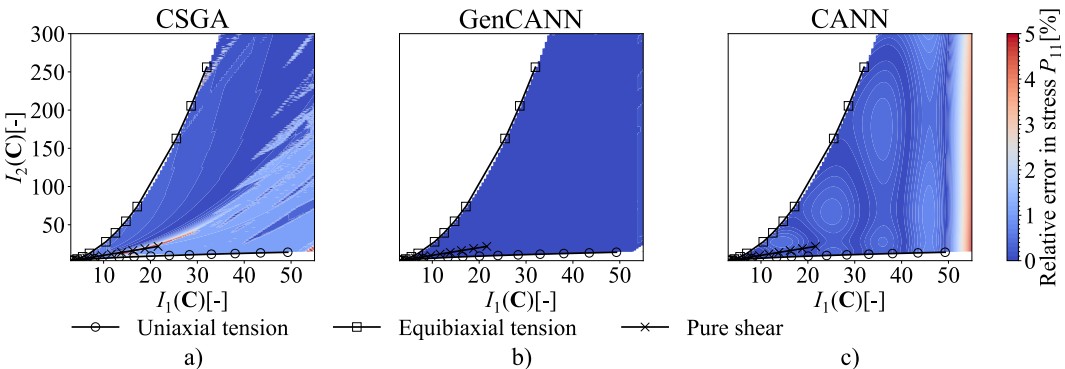

Figure 6: Predictions of the LLM-generated constitutive artificial neural network (GenCANN) for mechanical stress induced by deformation of a fictitious rubber-like material evaluated on a plane of biaxial loading states. The three marked paths are included in the training set, while all intermediate states are unseen. The GenCANN is compared against two benchmarks: the LLM-based constitutive scientific generative agent (CSGA) and the human-designed CANN.

Figures 4 and 5 show that GenCANN and CANN match measured and ground truth stresses with near-perfect accuracy across uniaxial, equibiaxial, and pure shear loading, while CSGA lags behind. These results confirm that CANNs, whether LLM-generated or manually implemented, outperform the unconstrained CSGA even on loading conditions known from training. We next evaluate model extrapolation to unseen loading scenarios. For the synthetic material, ground truth stresses can be computed for arbitrary deformations. This enables evaluation on Treloar's invariant plane (G.Treloar, 2005) (Figure 6), with the first and second invariants spanning the x- and y-axes, respectively. The plane spans from uniaxial to equibiaxial tension, with pure shear at the angular midpoint (note that the axes are scaled differently). Both the benchmarks CANN and CSGA extrapolate well to the new loading states between the marked paths. The CSGA shows slightly higher errors overall, but extrapolates better to the largest stretches. GenCANN excels in both generalization and extrapolation, performing remarkably well on loading conditions outside its training range.

### 4.3 SKIN DATA

To move beyond isotropy, we next study a transversely isotropic soft tissue: porcine skin. Aligned collagen creates one preferred fiber direction, with higher stiffness along the fibers and greater compliance across them. We use a publicly available biaxial stress–stretch dataset

with 402 data points from porcine skin specimens (Tac et al., 2022b;a). The five loading paths are equibiaxial, which applies equal stretch in both principal directions, strip-axial, which stretches one direction while keeping the other at its initial length, and off-axial, which stretches both directions with a stronger bias toward one. The applied stretch was increased monotonically during each test. We assume the tissue is incompressible and that there is no stress acting through the thickness. As a benchmark, we use the CANN variant selected through a systematic hyperparameter search in its original study (Linka et al., 2023b). Unlike our approach, this baseline model was trained on all available data points without a dedicated test split. The CSGA has so far only been implemented for isotropic materials, which is why it cannot serve as a benchmark for this dataset.

In contrast to uniaxial tension tests, biaxial tension tests report stresses in both in-plane directions. This provides directional information that uni-

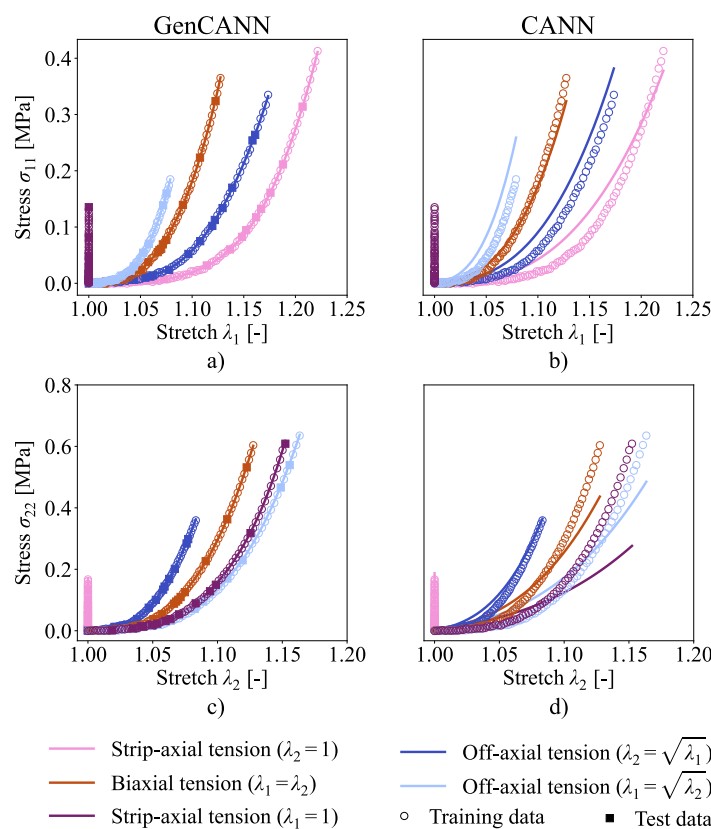

Figure 7: Predictions of the LLM-generated constitutive artificial neural network (GenCANN) for mechanical stress induced by porcine skin deformation compared against the human-designed CANN serving as benchmark.

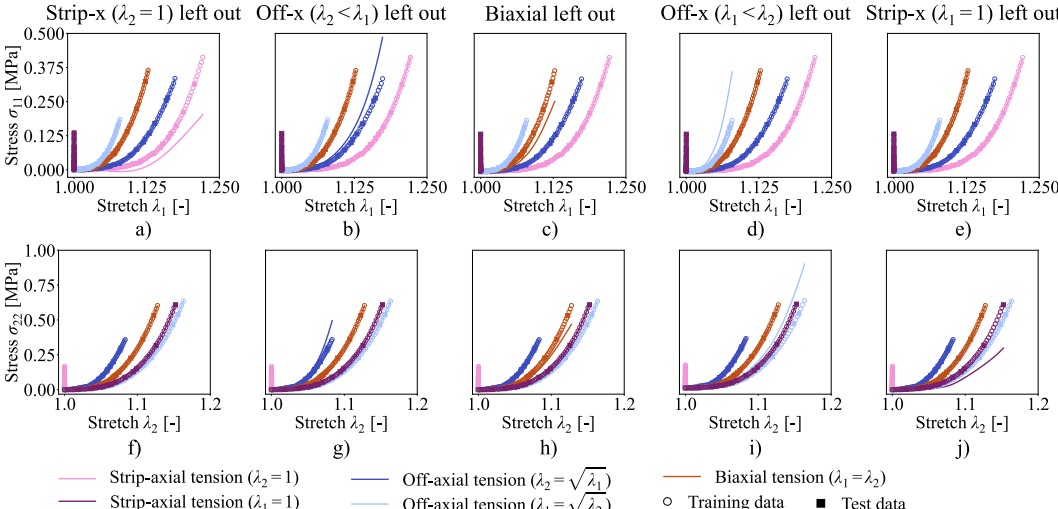

Figure 8: Predictions of the LLM-generated constitutive artificial neural network (GenCANN) for mechanical stress induced by porcine skin deformation evaluated using a leave-one-loading-scenario-out cross-validation. In five separate training runs, one loading path is excluded each time and used to test how well the GenCANN generalizes to unseen loading paths.

axial tests cannot capture and helps the models learn the fiber-induced anisotropy. On this dataset, the GenCANN fits all five loading paths with essentially perfect accuracy, see Figure 7. It reaches an $R^2$ score of 1 for every reported stress component. The manually implemented CANN by Linka et al. (2023b) shows noticeable errors even on loading paths included in the training, with $R^2$ scores such as 0.92-0.93 for equibiaxial loading. To check for overfitting, we used leave-one-loading-scenario-out cross-validation, retraining our GenCANN five times and evaluating its performance on the left-out path. As shown in Figure 8, predictions on unseen paths are less accurate than on paths known from training but remain on par with the manually engineered CANN even though that model was trained on all paths, indicating that the GenCANN does not overfit and extrapolates reasonably well to new biaxial loading states.

### 4.4 STATISTICAL ANALYSIS

In Section 3, we described the CANN generation pipeline, and Figure 9 summarizes this workflow and the step-wise statistics. The LLM designs a CANN, which is then dynamically executed, trained, and evaluated. If the code is not valid (syntax error) or training produces a negative $R^2$ score (training error), the script and error message are sent back to the LLM for correction. Across 124 individual implementations, 31% had syntax errors and 5% had training errors, both resolved by retries. Once a valid model is obtained, we return the script with its $R^2$ score for three refinement rounds. The first-iteration models appear already useful, and the refinement yields a small but consistent gain in accuracy and reduced variance, improving trustworthiness. To assess stability, we repeated the full generation five times per dataset and report the $R^2$ score averaged over loading scenarios in the lowest bar plot of Figure 9. For both rubber datasets, all runs reached near-perfect accuracy, resulting in no variance. For the brain and skin dataset, the variance is small and the mean $R^2$ scores remain well above 0.9. The best run for each dataset is highlighted in pink and its predictions appear in the figures. Because the process is partly stochastic, we recommend running multiple generations and selecting the best model, as we do here.

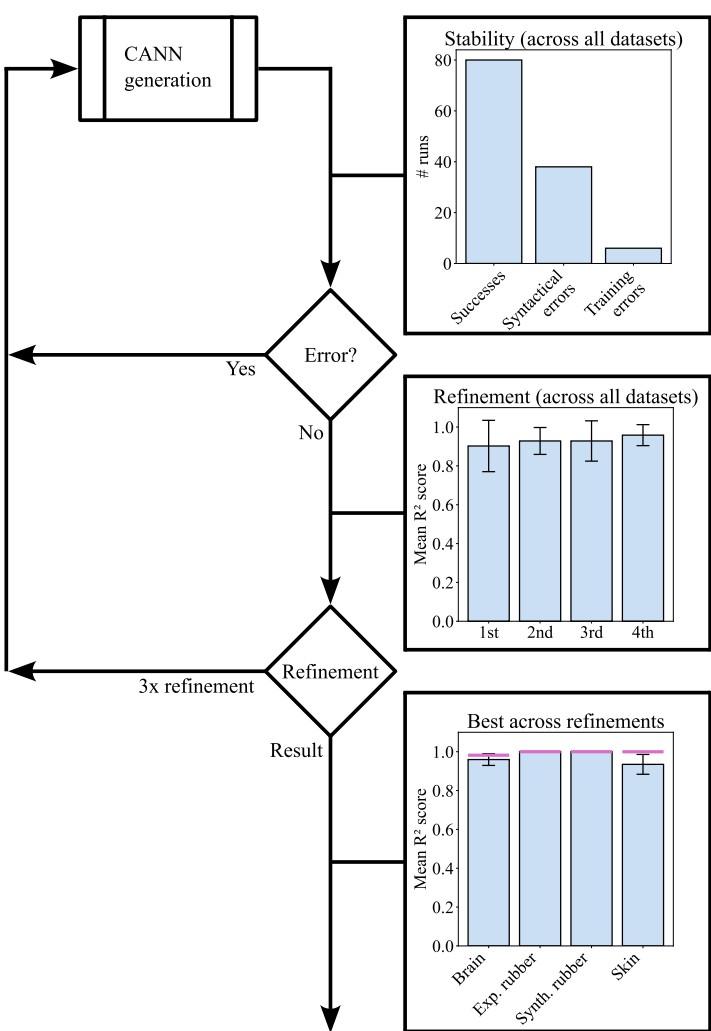

Figure 9: Stability analysis of our framework. If a generated CANN is invalid Python code (syntactical error) or yields a negative $R^2$ score (training error), we discard it and repeat the generation. After obtaining a valid CANN, we resend it three times with its $R^2$ score to the LLM for refinement. We repeat the full process five times per dataset, summarize the results in the lowest bar plot, and mark the best run in pink. The consistently accurate outcomes across runs confirm the stability of our approach.

## 5 DISCUSSION AND CONCLUSIONS

We were inspired by works such as (Wang et al.; Chen et al., 2024b) that create agents for use in LLM-based frameworks on demand. We applied this idea to constitutive modeling by generating specialized constitutive artificial neural networks (CANNs) on demand, each tailored to a specific material. Instead of viewing LLM-based approaches and specialized methods as competing, we propose to integrate them. Our approach combines the strengths of both. CANNs provide high accuracy and strict adherence to physical constraints, while LLMs offer an accessible interface and great flexibility while vastly reducing the human expertise required.

In detail, our framework automates constitutive modeling by prompting an LLM to design a CANN that fits the material class and data. The LLM makes all key design choices, including architecture, activation functions, constraints, fiber direction handling, and the full technical implementation. This gives the system high flexibility for modeling new materials. Static code manages prompt selection and model training, which reduces user effort but limits adaptability. User input is minimal, requiring only material classification and data, making the system both powerful and easy to use. In the future, automating these static parts with LLM agents could improve flexibility and user experience even further.

The LLM-generated CANNs (GenCANNs) matched or, in several cases, clearly exceeded the accuracy of human-designed CANNs across the brain, rubber, and skin datasets, supporting the viability of our approach. Among the LLM's design choices, we observed a consistent preference for larger feedforward architectures than the baselines, for example, 256–128–64–3 vs. 100 neurons for brain and 32–32 vs. 16–16 for synthetic rubber (see Table 2), which raises the question of whether the performance gains are due only to increased capacity or whether GenCANNs remain competitive when restricted to the same size as the baselines. To answer this, we repeated the experiments with GenCANNs constrained to exactly match the baseline CANN network sizes. For brain and rubber, the constrained and unconstrained GenCANNs performed indistinguishably. Only the skin dataset showed a benefit from the larger, unconstrained network on the training paths, yet even there, the constrained GenCANN still clearly surpassed the baseline CANN. While larger models can fit training data more closely, they also increase the risk of overfitting, especially with the small datasets typical of constitutive modeling (e.g., 15 data points per loading path in rubber). Our generalization tests indicate that the LLM-based design remains well balanced: on the invariant plane and in the leave-one-loading-scenario-out cross-validation for skin, both constrained and unconstrained Gen-CANNs generalize to unseen loading states and extrapolate beyond the trained range with remarkable accuracy. The full analysis of the network size, including all plots, is provided in Section A.1 and shows that GenCANNs remain highly competitive even when constrained in size. Our goal is to simplify the generation of CANNs for new materials rather than to reproduce or beat existing manually designed models. Overall, our results show that LLM-generated, physics-constrained CANNs are ready for real-world applications.

Future work could extend this paradigm of using LLMs to generate specialized modules on demand for tasks beyond constitutive modeling. Another direction is to deepen the integration between CANNs and LLMs within constitutive modeling. Components that are currently static, such as prompt selection by material class and the orchestration of training and evaluation, could be assigned to LLM-driven agents. This would expand the design space, reduce manual intervention, and improve adaptability to new materials and evolving model requirements.

## Ethics statement

All datasets used in this study, including those on the deformation of human brain tissue and porcine skin tissue under mechanical load, were taken from previously published literature. No new experiments involving human or animal tissues were conducted specifically for this study.

## Reproducibility statement

The human brain tissue and porcine skin tissue datasets, along with the corresponding CANN code, are available at: https://github.com/LivingMatterLab/CANN. The rubber datasets and the corresponding CANN implementation are available in this repository: https://github.com/ConstitutiveANN/CANN. The implementation of the CSGA can be found here: https://github.com/ConstitutiveSGA/CSGA. Finally, the GenCANN code is available at: https://github.com/gencann25/GenCANN.

## LLM usage

Besides the obvious research on LLMs, their use in this work was limited to refining the wording of a few sentences in the manuscript. All such LLM-assisted formulations were carefully reviewed by the authors, who take full responsibility for the entire manuscript.

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

# A APPENDIX

## A.1 NETWORK ARCHITECTURE EVALUATION

In Section 3, we described the LLM's design space when generating a CANN. In Section 4, we showed that these GenCANNs achieve very high accuracy. While preprocessing from the deformation gradient to invariants and postprocessing from strain energy to stresses follow established continuum mechanics, the LLM still makes several important choices: which invariant combinations to use, the network architecture and size, activation functions, constraints and regularization, how to handle or estimate fiber directions, and weight initialization. We observed that it often selects larger architectures than the baseline CANNs (Table 2). To isolate the effect of capacity from other design choices, we ran a controlled comparison. Each GenCANN was constrained to use exactly the baseline CANN network size and evaluated alongside the unconstrained GenCANNs and the baseline CANNs (Figures 10–15).

For the brain data (Figure 10) and the rubber data on trained loading paths (Figures 11 and 12), the three models are indistinguishable in practice. When we evaluate generalization on the synthetic rubber material using Treloar's invariant plane (Figure 13), both GenCANNs, constrained and unconstrained, reach a remarkably high accuracy and substantially outperform the baseline CANN across unseen mixed biaxial states. For skin (Figure 14), the unconstrained GenCANN is the only model that perfectly fits all training paths, suggesting that the larger architecture helps for this more complex anisotropic case, yet the constrained GenCANN still outperforms the baseline. In the skin leave-one-loading-scenario-out cross-validation (Figure 15), the constrained GenCANN generalizes at least as well as the unconstrained model to the left-out path; the unconstrained model fits training paths more tightly but does not generalize better.

We consider the unconstrained GenCANNs the most realistic choice for new materials, where no manually tuned baseline prescribes an architecture. Our aim is to remove manual trial-and-error, not to reproduce legacy sizes. Still, when we do restrict the LLM to baseline sizes, GenCANNs remain highly competitive: only the skin dataset clearly benefits from the larger network, and even there the constrained GenCANN exceeds the baseline. For generalization to unseen states and extrapolation (Figures 13 and 15), constrained GenCANNs are on par with unconstrained ones. Overall, the strong performance of LLM-designed CANNs cannot be attributed to network size alone, and if users prefer smaller models for efficiency, interpretability, or deployment constraints, the GenCANN approach can honor those limits while maintaining high accuracy.

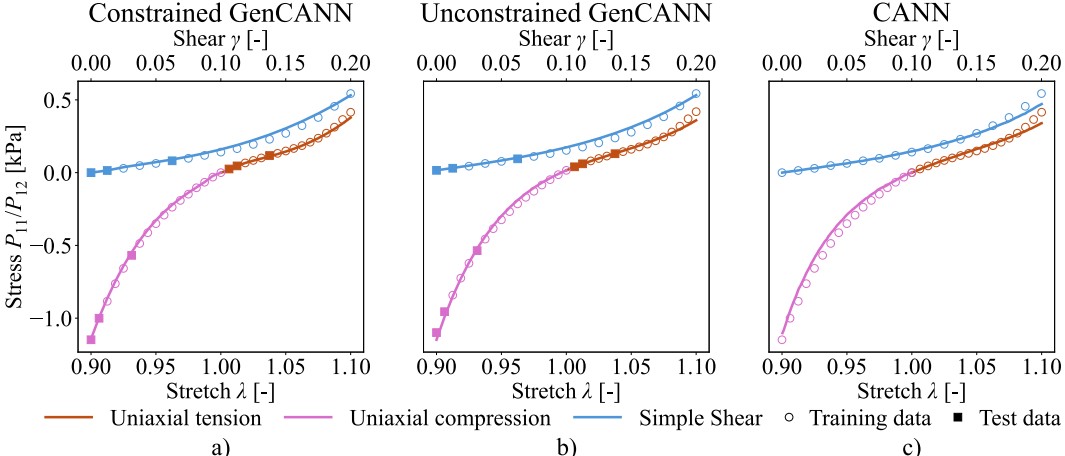

Figure 10: Comparison of the LLM-generated constitutive artificial neural network (GenCANN) constrained to the same network size as the baseline CANN with this baseline CANN and the unconstrained GenCANN on experimental brain data. All models reach performance close to ideal.

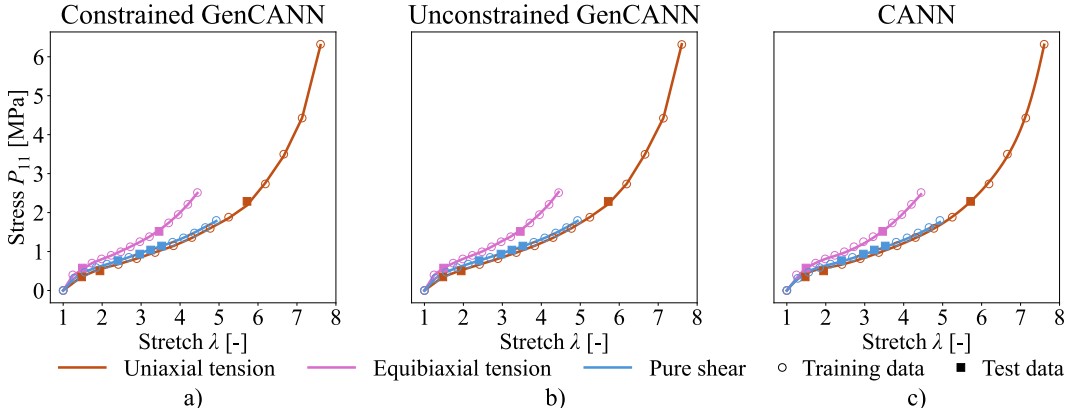

Figure 11: Comparison of the LLM-generated constitutive artificial neural network (GenCANN) constrained to the same network size as the baseline CANN with this baseline CANN and the unconstrained GenCANN on experimental rubber data. All models reach performance close to ideal.

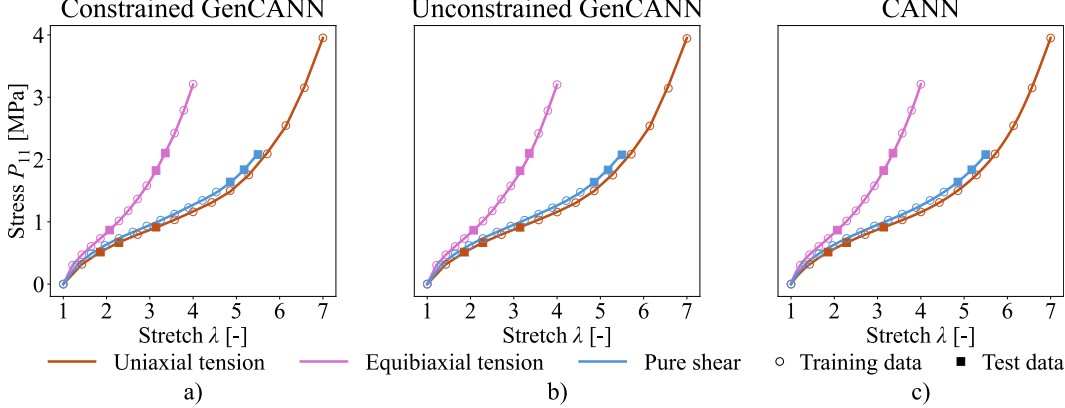

Figure 12: Comparison of the LLM-generated constitutive artificial neural network (GenCANN) constrained to the same network size as the baseline CANN with this baseline CANN and the unconstrained GenCANN on synthetic rubber data. All models reach performance close to ideal.

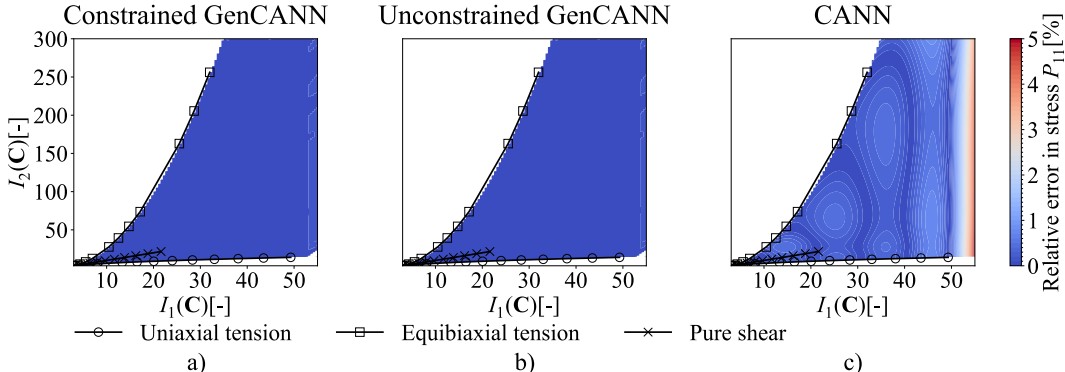

Figure 13: Comparison of the LLM-generated constitutive artificial neural network (GenCANN) constrained to the same network size as the baseline CANN with this baseline CANN and the unconstrained GenCANN on the synthetic rubber invariant plane. Both GenCANN variants generalize far better than the baseline CANN, with only minor differences in performance.

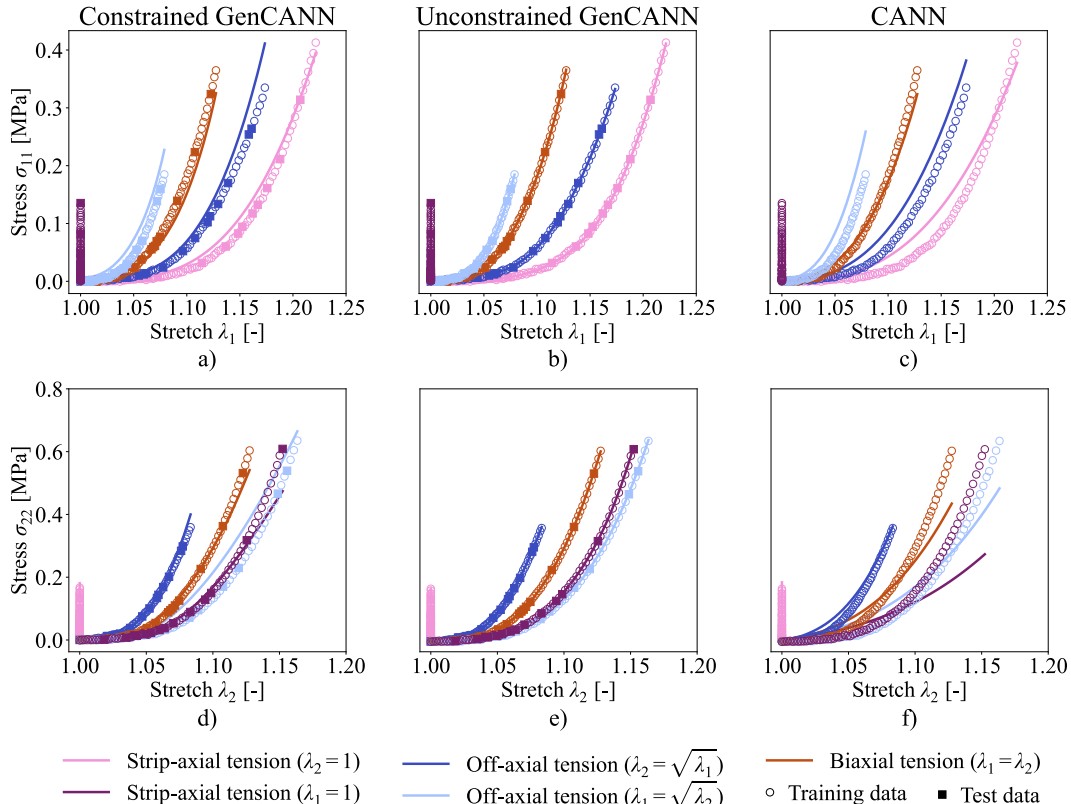

Figure 14: Comparison of the LLM-generated constitutive artificial neural network (GenCANN) constrained to the same network size as the baseline CANN with this baseline CANN and the previously presented unconstrained GenCANN on experimental skin data. The unconstrained GenCANN shows the highest performance, with the constrained GenCANN clearly ahead of the baseline CANN.

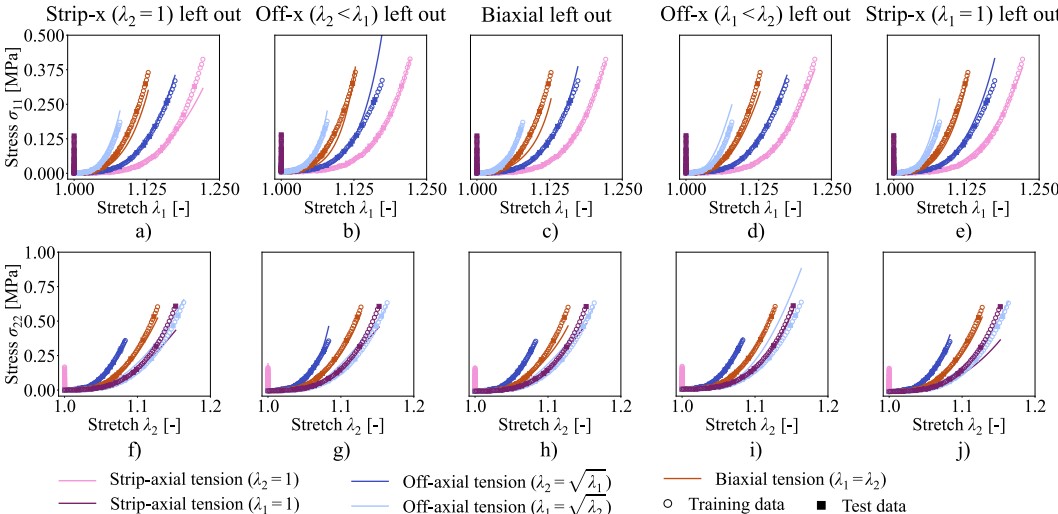

Figure 15: Predictions of the LLM-generated constitutive artificial neural network (GenCANN) constrained to the same network size as the baseline CANN for mechanical stress induced by porcine skin deformation evaluated using a leave-one-loading-scenario-out cross-validation. Compared to Figure 8, the constrained GenCANN generalizes to unseen loading scenarios on par with the unconstrained model.

## A.2 TABLES

| Test | CSGA | Constrained GenCANN | Unconstrained GenCANN | CANN |
|------|------|------|------|------|
| **Brain** | | | | |
| Uniaxial tension | 0.93 | 0.99 | 0.97 | 0.96 |
| Uniaxial compression | 1.00 | 1.00 | 1.00 | 0.99 |
| Simple Shear | 0.99 | 0.99 | 0.98 | 1.00 |
| **Experimental rubber** | | | | |
| Uniaxial tension | 0.99 | 1.00 | 1.00 | 1.00 |
| Equibiaxial tension | 0.97 | 1.00 | 1.00 | 1.00 |
| Pure shear | 0.98 | 1.00 | 1.00 | 1.00 |
| **Synthetic rubber** | | | | |
| Uniaxial tension | 0.87 | 1.00 | 1.00 | 1.00 |
| Equibiaxial tension | 0.98 | 1.00 | 1.00 | 1.00 |
| Pure shear | 0.93 | 1.00 | 1.00 | 1.00 |
| **Skin** | | | | |
| Strip-X ($\lambda_2 = 1$), stress in 1 | | 0.98 | 1.00 | 0.96 |
| Strip-X ($\lambda_2 = 1$), stress in 2 | | 0.94 | 1.00 | 0.98 |
| Off-X ($\lambda_2 = \sqrt{\lambda_1}$), stress in 1 | | 0.93 | 1.00 | 0.97 |
| Off-X ($\lambda_2 = \sqrt{\lambda_1}$), stress in 2 | | 0.99 | 1.00 | 0.95 |
| Equibiaxial ($\lambda_1 = \lambda_2$), stress in 1 | | 0.97 | 1.00 | 0.92 |
| Equibiaxial ($\lambda_1 = \lambda_2$), stress in 2 | | 0.99 | 1.00 | 0.93 |
| Off-X ($\lambda_1 = \sqrt{\lambda_2}$), stress in 1 | | 0.79 | 1.00 | 0.89 |
| Off-X ($\lambda_1 = \sqrt{\lambda_2}$), stress in 2 | | 0.89 | 1.00 | 0.90 |
| Strip-X ($\lambda_1 = 1$), stress in 1 | | 0.87 | 1.00 | 0.26 |
| Strip-X ($\lambda_1 = 1$), stress in 2 | | 0.96 | 1.00 | 0.81 |

Table 1: $R^2$ scores of the constitutive scientific generative agent (CSGA), the constitutive artificial neural network (CANN), the LLM-generated CANN (GenCANN), and the GenCANN constrained to the baseline CANN network size, as shown in Figures 3-8 and 10-15.

| Material | Model | Neurons per hidden layer |
|---|---|---|
| Brain | CANN | 100 |
| | Constrained GenCANN | 100 |
| | Unconstrained GenCANN | 256, 128, 64, 3 |
| Experimental rubber | CANN | 16, 16 |
| | Constrained GenCANN | 16, 16 |
| | Unconstrained GenCANN | 64, 64, 16, 16 |
| Synthetic rubber | CANN | 16, 16 |
| | Constrained GenCANN | 16, 16 |
| | Unconstrained GenCANN | 32, 32 |
| Skin | CANN | 8, 16 |
| | Constrained GenCANN | 8, 16 |
| | Unconstrained GenCANN | 128, 128, 64, 32 |

Table 2: Architectural specifications of the constitutive artificial neural network (CANN), the LLM-generated CANN (GenCANN), and the GenCANN constrained to the baseline CANN network size, as shown in Figures 3-8 and 10-15.

### A.3 Exemplary GenCANN implementation

Listing 1: Full implementation of LLM-generated constitutive artificial neural network (GenCANN) for skin dataset.

```python
import numpy as np
import tensorflow as tf

# ---------------------------------------#
#               Helper layer              #
# ---------------------------------------#
class StructureTensor(tf.keras.layers.Layer):
    """
    Learns the fibre angle alpha  (0  alpha  Pi/2) and returns the
    structure tensor
    N = n  n with  n = [cos alpha, sin alpha, 0]T.
    """
    def __init__(self, **kwargs):
        super().__init__(**kwargs)
        self.alpha = self.add_weight(
            name        = "alpha",
            shape       = (),
            dtype       = tf.float32,
            initializer = tf.keras.initializers.RandomUniform(0.0, 0.5*np
.pi),
            trainable   = True,
        )

    def call(self, F):
        batch = tf.shape(F)[0]
        n = tf.stack(
            [tf.cos(self.alpha), tf.sin(self.alpha), tf.constant(0.0,
dtype=F.dtype)],
            axis=0,
        )                                          # (3,)
        n = tf.tile(n[None, :], [batch, 1])    # (B,3)
        return tf.einsum("bi,bj->bij", n, n)   # (B,3,3)

# ---------------------------------------#
#               Psi-network               #
# ---------------------------------------#
class PsiNetwork(tf.keras.layers.Layer):
    """
    A single multivariate neural network Psi(lambdax, lambday, lambdaz, I
, II, IV, V).
    Feeding both the stretches and the invariants makes it easier for the
 NN
    to discover patterns for extreme stretch states (strip-y, off-y, ).
    """
    def __init__(self, **kwargs):
        super().__init__(**kwargs)
        self.net = tf.keras.Sequential([
            tf.keras.layers.Dense(128, activation="softplus"),
            tf.keras.layers.Dense(128, activation="softplus"),
            tf.keras.layers.Dense(64 , activation="softplus"),
            tf.keras.layers.Dense(32 , activation="softplus"),
            tf.keras.layers.Dense(1  , activation="linear"),
        ])

    def call(self, features):
        return self.net(features)    # (B,1)

# ---------------------------------------#
```

```python
# -------------------------------------------------------------------------#
# CANN                                                                     #
# -------------------------------------------------------------------------#
class CANN(tf.keras.layers.Layer):
    """
    Constitutive Artificial Neural Network for an incompressible, transversely
    isotropic material subjected to planar stretches.
    """
    def __init__(self, **kwargs):
        super().__init__(**kwargs)
        self.structure_tensor = StructureTensor(name="structure_tensor")
        self.psi_network      = PsiNetwork     (name="psi_network")

    # ----------------------------------------------------------------#
    #                   Public forward pass                          #
    # ----------------------------------------------------------------#
    def call(self, inputs):
        """
        Args:
            inputs (list | tuple): (stretch_x, stretch_y)  two tensors of
                                    shape (B,) and dtype float32.

        Returns:
            list(tf.Tensor, tf.Tensor): sigma_xx and sigma_yy (each shape (B,))
        """
        if not isinstance(inputs, (list, tuple)) or len(inputs) != 2:
            raise ValueError("CANN expects [stretch_x, stretch_y] as input.")

        lambdax = tf.reshape(inputs[0], (-1,))
        lambday = tf.reshape(inputs[1], (-1,))
        lambdaz = 1.0 / (lambdax * lambday)        # incompressibility J = 1

        # (B,3,3) deformation gradient  (diagonal for pure stretches)
        F = tf.linalg.diag(tf.stack([lambdax, lambday, lambdaz], axis=1))

        # Isochoric part of the first PK stress
        P_iso = self._compute_P_iso(F, lambdax, lambday, lambdaz)

        # Convert to Cauchy stress and eliminate pressure
        sigma = self._compute_cauchy(F, P_iso)         # (B,3,3)

        return [sigma[:, 0, 0], sigma[:, 1, 1]]

    # ----------------------------------------------------------------#
    #                   Internal helpers                             #
    # ----------------------------------------------------------------#
    def _compute_P_iso(self, F, lambdax, lambday, lambdaz):
        """
        Obtain P_iso = Psi/F  via automatic differentiation.
        Feature vector for Psi contains the stretches AND four invariants.
        """
        with tf.GradientTape() as tape:
            tape.watch(F)

            # Right Cauchy-Green tensor
            C   = tf.matmul(tf.transpose(F, [0, 2, 1]), F) # (B,3,3)
            trC = tf.linalg.trace(C)                       # (B,)
            trC2 = tf.linalg.trace(tf.matmul(C, C))        # (B,)

            I1 = trC[:, None]                              # (B,1)
            I2 = (0.5 * (trC ** 2 - trC2))[:, None]        # (B,1)
```

```
115
116            N  = self.structure_tensor(F)
117            IV = tf.reduce_sum(C * N,            axis=[-2, -1])[:, None
      ]  # (B,1)
118            V  = tf.reduce_sum(tf.matmul(C, C) * N, axis=[-2, -1])[:,
      None] # (B,1)
119
120            stretches = tf.stack([lambdax - 1.0, lambday - 1.0, lambdaz -
       1.0], axis=1)      # (B,3)
121            invariants = tf.concat(
122                [I1 - 3.0, I2 - 3.0, IV - 1.0, V - 1.0], axis=1
123            ) # (B,4)
124
125            features = tf.concat([stretches, invariants], axis=1) # (B,7)
126
127            Psi = self.psi_network(features) # (B,1)
128
129        return tape.gradient(Psi, F)    # (B,3,3)
130
131    def _compute_cauchy(self, F, P_iso):
132        """
133        sigma = (1/J) * P_iso * FT    p I,  with p chosen such that
      sigma_zz = 0.
134        """
135        J  = tf.linalg.det(F)  # (B,)
136        sigma = tf.matmul(P_iso, tf.transpose(F, [0, 2, 1])) / J[:, None,
       None] # (B,3,3)
137
138        p  = sigma[:, 2, 2]  # (B,)
139        I3 = tf.eye(3, batch_shape=[tf.shape(F)[0]], dtype=F.dtype)
140        return sigma - p[:, None, None] * I3
141
142    # ---------------------------------------#
143    #           Public getter               #
144    # ---------------------------------------#
145    def get_alpha(self):
146        """Return the learned fibre direction angle alpha (radians)."""
147        return self.structure_tensor.alpha
148
149 # ---------------------------------------#
150 #           Model builder               #
151 # ---------------------------------------#
152 def build_cann_model():
153     """
154     Creates the Keras model that maps planar stretches to normal Cauchy
      stresses.
155
156     Returns:
157         tf.keras.Model
158     """
159     stretch_x = tf.keras.Input(shape=(), name="stretch_x")
160     stretch_y = tf.keras.Input(shape=(), name="stretch_y")
161
162     sigmax, sigmay = CANN(name="cann")([stretch_x, stretch_y])
163
164     model = tf.keras.Model(
165         inputs=[stretch_x, stretch_y],
166         outputs=[sigmax, sigmay],
167         name="CANN_model",
168     )
169     return model
```

