# OpenReview forum: "Automating modeling in mechanics: LLMs as designers of physics-constrained neural networks for constitutive modeling of materials"
_ICLR.cc/2026/Conference — Submitted to ICLR 2026_

### Official Review · Reviewer_cWjb · 2025-10-19

**Soundness:** 1
**Presentation:** 2
**Contribution:** 1
**Rating:** 2
**Confidence:** 3

**Summary:**

This paper presents GenCANN, a framework that leverages LLMs to automatically design CANNs for modeling the mechanical behavior of materials. By combining CANNs with the automation and flexibility of LLM-based code generation, the approach enables on-demand creation of physics-constrained neural networks tailored to specific materials and datasets. Experiments show that the LLM-generated models match or exceed the accuracy and generalization performance of manually engineered CANNs, demonstrating the potential of LLMs as autonomous designers of scientific, physics-informed machine learning models.

**Strengths:**

- This paper utilizes LLM to automatically design CANNs for modeling the mechanical behavior of materials.

- The authors evaluate their approach on three representative benchmarks (brain, rubber, and skin).

**Weaknesses:**

- The technical contribution is limited. The work directly integrates two existing ideas (CANNs and LLM-based code generation). There is no deeper theoretical exploration or experimental analysis of the system (physical consistency, generation success rate, failure cases, …). The writing reads more like an engineering report than a scientific research paper.

- The experimental scope is narrow, being restricted to hyperelastic materials. This significantly underutilizes the potential breadth of LLM-driven scientific modeling. Broder scenarios should be included to justify the value and meaning.

- According to the method section, the LLM seems only to be responsible for generating the CANN architecture, while other parts such as data preparation and training still remain manually configured. This makes the system resemble a “customized API” rather than a genuinely autonomous agentic framework.

**Questions:**

- All examples in the paper are limited to simple MLP-based architectures. The LLM appears to only determine the number of layers and neurons. Can the framework generate architectures with more structural diversity or physical inductive bias (e.g., CNNs, GNNs, Transformers or operator networks)?

- Regarding MLP generation, many existing techniques such as neural architecture search (NAS) and AutoML frameworks can already perform automatic architecture optimization. What are the concrete advantages of using an LLM over these established approaches in this specific context? The theoratical and experimental comparisons are necessary.

- From the Table 2, all GenCANNs have more neurons than CANNs.  Does this mean that GenCANN just generates MLPs with more parameters rather than design the architecture reasonably?

---

> ### Author Response · Authors · 2025-11-24
> **Comment**
>
> We appreciate your review and your constructive feedback on our manuscript. Below, we address the weaknesses and questions you raised point by point:
> - You note the lack of deeper experimental analysis of the system. To address this, we have substantially expanded our evaluation of the CANN generation process. The revised manuscript now includes an examination of error modes during generation, a discussion of error-handling strategies, statistical evaluation of the refinement steps, and a quantitative assessment of model accuracy across multiple runs. These additions provide insights that go beyond our specific use case and contribute to a broader understanding of how LLMs can be integrated into scientific code generation workflows.
> - You ask whether our framework can generate architectures that are not based on MLPs. In principle, the approach could certainly be extended to produce different architectures. However, in the current work, we have focused exclusively on constitutive modeling through CANNs, which are inherently MLP-based.
> - You ask about the advantages of using an LLM compared to existing automatic architecture optimization techniques such as NAS or AutoML. We would like to clarify that CANNs go far beyond simple MLP design. Their architectures explicitly mirror continuum mechanics theory and guarantee adherence to physical constraints such as objectivity and thermodynamic consistency. Implementing a CANN involves a series of highly specialized design decisions, including the choice of invariant combinations, network architecture and size, activation functions, constraints and regularization, handling and estimation of fiber directions when required, weight initialization, and treatment of boundary conditions. While some of these aspects relate to the MLP core, many are unique to this application and cannot be addressed by standard architecture optimizers.
> For this reason, a direct comparison of advantages and disadvantages is not meaningful - these approaches do not compete in the same space. Our framework is designed to automate the creation of physics-constrained neural networks for constitutive modeling, not generic architecture search.
> - Your last question refers to the number of hidden layers and neurons in GenCANN networks compared to CANN networks. We recognize that this difference in hyperparameters is important and have addressed it by conducting a new set of experiments where the LLM was constrained to match the baseline architecture. These results, now detailed in Section A1 with six supporting plots and centrally discussed in Section 5, show that for brain and rubber datasets, constrained and unconstrained GenCANNs perform with similar accuracy. For skin, the unconstrained model benefits from its larger size, yet even the constrained GenCANN still clearly outperforms the baseline. While larger networks can fit training data more closely, they also risk overfitting - especially with small datasets typical in constitutive modeling. Our generalization tests confirm that both constrained and unconstrained GenCANNs extrapolate beyond the training range and generalize to unseen loading states with remarkable accuracy.

---

### Official Review · Reviewer_JyT9 · 2025-10-25

**Soundness:** 3
**Presentation:** 4
**Contribution:** 3
**Rating:** 8
**Confidence:** 4

**Summary:**

This paper introduces a novel framework for automating constitutive modeling in solid mechanics. The authors propose using an LLM to generate the complete code for a specialized, physics-constrained neural network architecture known as a Constitutive Artificial Neural Network (CANN). The proposed GenCANN approach is tailored on-demand to a specific material class and dataset. The framework combines easy-to-use LLM-based systems with highly accurate, physically consistent CANNs. The authors demonstrate their approach on three real-world experimental datasets comprising human brain tissue, rubber, and porcine skin. The experiments show that the LLM-generated GenCANNs achieve performance that is on par with or better than CANNs, while clearly outperforming a previous LLM-based approach.

**Strengths:**

- The core contribution is simple, elegant, and highly effective. Letting an LLM generate a specialized, physics-informed model is a powerful paradigm. This approach leverages the LLM’s strength in code generation, and combines it with existing research on constitutive modeling techniques.
- The paper is really well-written and easy to follow. The motivation is clear, the background is explained concisely, and the proposed method is presented logically. The figures are informative and effectively support the results. The authors accurately and concisely position their work within the existing literature and highlight the unique novelty and contribution of their method.
- The paper is supported by a strong set of experiments on three distinct and challenging real-world datasets. The evaluation includes extrapolation to unseen invariants and stretches. In all experiments, the proposed model is on par with or better than existing CANNs, while being easier to handle. Similarly, it is significantly better than the LLM-based CSGA baseline, which has similar complexity for the user.

**Weaknesses:**

- According to Table 2, the GenCANN architectures are consistently and significantly larger (i.e., more neurons and/or layers) than their manually-designed counterparts. For instance, the baseline CANN for skin has "12, 12" neurons per layer, while the GenCANN has "128, 128, 64, 32". This raises the question of whether the superior performance of GenCANN is due to the LLM's intelligent design or simply its increased model capacity. While the core contribution of automatically generating CANNs is not affected by this, the larger model architecture seems like a potentially unfair advantage of the GenCANN model.
- The LLM design process is a bit opaque, and the risks and failure modes that are associated with it are not investigated. The paper successfully shows that an LLM can generate a high-performing CANN, but it does not explore cases where the LLM behaves unexpectedly or fully fails. Especially with complex LLM-generated code, such as the example CANN provided in the appendix, these failure modes are the be expected in some cases, and a way to deal with them would strengthen the method and its applicability.
- Related to the previous point, the paper mentions that the LLMs are given three iterations to refine their proposed CANN, using R2 score as the main feedback. Here, it would be interesting to see how much (or if) this refinement actually helps, especially since the R2 score is a pretty sparse metric that does not provide much feedback to the LLM.
- All experiments seem to be based on one random seed. A more rigorous scientific evaluation with more repetitions would validate the statistical significance of the method and, potentially, give insight into the frequency of failure for the LLM-generated solutions.

**Questions:**

- Could the authors please comment on the significant difference in architectural complexity between the GenCANNs and the baseline CANNs? How much of the observed performance improvement can be attributed to the LLM's design choices versus the fact that the GenCANNs are simply larger models? It would strengthen the paper to include results from a baseline CANN that is manually scaled up to a similar complexity as the GenCANN.
- The iterative refinement process is an interesting component of the proposed framework. Could the authors provide more detail on how the LLM improves its generated code over multiple rounds given only a single R2 score as feedback? An example showing the evolution of a generated CANN from one iteration to the next would be very illuminating. Similarly, have other iterative approaches been considered?
- What are the failure modes of this approach? Were there instances where the LLM generated code that was syntactically incorrect, did not adhere to the required physical constraints, or simply failed to train effectively? How common are these failures?
- In Figure 6, the CSGA baseline performs poorly on the pure shear data (which is in-distribution) but seems to extrapolate better than the baseline CANN in other regions. Failing in-distribution and generalizing well out-of-distribution is a pretty uncommon behavior. Is there an intuition for these specific results?
- In Figure 7, the test data points seem to be missing or mis-labelled (i.e., there are only circles denoting training data, no squares for the test data in the plots). Could the authors update this figure to better showcase the relation between training and test data?
- From the provided CANN example, it is not directly clear how large the CANN design space actually is. Can the authors provide a bit of clarification on the kind of decisions that need to be made by the LLM, maybe by highlighting the “interesting” choices in the generated example?

---

> ### Comment · Reviewer_JyT9 · 2025-11-13
> **Updated score**
>
> After reading the other reviewer's significantly more critical assessment of the provided submission, I agree with some of the weaknesses mentioned. The baselines' relatively small models may account for most of the provided performance gain, and its not clear from the evaluation how robust the results are. The results are also not entirely unexpected given the scope of the task and the known capabilities of LLMs. I still believe that the paper has an interesting idea, very solid writing and enough interesting insights to be relevant for the research community. Using single-seed evaluations is (unfortunately) pretty common in LLM research, and I believe that there is a benefit to providing "expected" results, even if the scientific novelty is limited. I will update my score to a "weak accept" to reflect that I still like the paper, but agree with the weaknesses pointed out by my fellow reviewers, and will thus not strongly argue for acceptance.

---

> ### Author Response · Authors · 2025-11-24
> **Comment 1/2**
>
> Thank you very much for your encouraging feedback! We address the weaknesses and questions you raised point by point below:
> - We appreciate your recognition of our core contribution, which is the automatic generation of CANNs. We also understand that the difference in network size between LLM-generated CANNs and manually designed CANNs is an important point that deserved more attention. Thank you for suggesting a comparison with manually scaled-up baseline CANNs. However, we would like to keep the baseline CANNs unchanged to preserve their role as expert-optimized, peer-reviewed models. Instead, we conducted a new set of experiments where the LLM was constrained to match the baseline architecture. These results, now detailed in Section A1 with six supporting plots and centrally discussed in Section 5, show that for brain and rubber datasets, constrained and unconstrained GenCANNs perform with similar accuracy. For skin, the unconstrained model benefits from its larger size, yet even the constrained GenCANN still clearly outperforms the baseline. While larger networks can fit training data more closely, they also risk overfitting - especially with small datasets typical in constitutive modeling. Our generalization tests confirm that both constrained and unconstrained GenCANNs extrapolate beyond the training range and generalize to unseen loading states with remarkable accuracy.
> - You raise an important point regarding the lack of detailed analysis of the LLM design process, including failure modes and the effectiveness of iterative refinement. We have now added a comprehensive evaluation of these aspects in Section 4.4 and summarize the results in Figure 9. In short, we identified two main failure modes: syntactical errors, where the generated Python code could not be executed, and training errors, where the model failed to capture the training data and produced an average R² score below zero across loading scenarios. Both issues were resolved by returning the implementation and a description of the error to the LLM for correction. Without error handling, syntactical errors occurred in 31% of test runs and training errors in 5%, but our error-handling pipeline, including the refinement process, successfully resolved all errors encountered during CANN generation.
> Regarding refinement efficacy, our statistical analysis of three refinement rounds across all datasets, with five runs per dataset, shows that the first generation already achieves high accuracy, while subsequent rounds provide small but significant improvements in accuracy and reductions in variance. For example, final results for the rubber dataset exhibit zero variance across runs, while brain and skin datasets show only minor variance. Because the pipeline is partly stochastic, we recommend running it multiple times and selecting the most accurate model, as we did. To illustrate the evolution of generated CANNs across iterations, we have uploaded the complete conversation logs for all runs and datasets to the GitHub repository, where they can be examined in detail.
> - You ask about the unusual behavior of the CSGA baseline, which performs poorly on pure shear data while generalizing better to other loading scenarios. We believe this can be explained by a closer look at the loading scenarios: the other loading scenarios used for training include only normal loading and no shear loading, and all intermediate scenarios represent interpolations between shear and normal loading. It appears that this specific CSGA model captures pure shear loading poorly but is able to represent pure normal loading and interpolations of shear and normal loading accurately enough to be useful.
> - Thank you for pointing out the issue with Figure 7. The baseline CANN does not use separate test data and is trained on all available data, which led us to mistakenly plot our GenCANN in the same way. However, as described in the manuscript, we did separate test data for evaluation. We have now updated Figure 7 to clearly distinguish training and test data.

---

> > ### Author Response · Authors · 2025-11-24
> > **Comment 2/2**
> >
> > - You note that the design space for LLM-generated CANNs is unclear from the provided CANN example. We agree that this design space is a crucial aspect of our work. However, we find it difficult to separate “interesting” from “non-interesting” design decisions, as nearly every line of generated code - beyond class and method signatures - reflects a choice made by the LLM. Highlighting all of these would not be particularly helpful.
> > To clarify, we have expanded our explanation of the LLM design space in Section 3. Additionally, Figure 2 illustrates its breadth. Implementing a CANN involves a series of highly specialized decisions, including the choice of invariant combinations, network architecture and size, activation functions, constraints and regularization, handling and estimation of fiber directions when required, weight initialization, and treatment of boundary conditions. All of these decisions are made by the LLM. We hope the revised Section 3 and Figure 2 make this clear.

---

### Official Review · Reviewer_GdJT · 2025-10-28

**Soundness:** 1
**Presentation:** 3
**Contribution:** 1
**Rating:** 2
**Confidence:** 4

**Summary:**

The submitted work, *GenCANN*, proposes a framework in which an LLM automatically generates a physics-constrained neural network for constitutive modelling. Specifically, the framework provides a template for constructing Constitutive Artificial Neural Networks (CANNs) tailored to user-specified materials and datasets. The authors evaluate the LLM-generated code on hyperelastic deformation problems involving rubber, skin, and brain tissue under various loading conditions. Results are compared against both a manually designed CANN and a Scientific Generative Agent adapted for constitutive modelling (CSGA, Tacke et al., 2025).

**Strengths:**

1. The paper is very well written and clearly structured, making it easy to read and follow.
2. The experimental section is comprehensive, covering multiple material types (e.g., skin, rubber) and including both real and synthetic datasets. The authors also test generalisation to unseen loading conditions and visualise the results neatly.

**Weaknesses:**

Despite the clarity and reasonable set of experiments, the paper has several major weaknesses regarding its _significance_ and _originality_.
 1. Limited contribution

The core contribution appears minimal. The authors essentially provide a prompt template for the LLM to generate code implementing a CANN. The method section is very brief (roughly half a page) and mainly describes the two-part LLM prompt (summary of continuum mechanics + skeleton code).  However, this lacks genuine research insight or technical contributions. There is no clear takeaway from the experiments beyond demonstrating that the LLM can replicate existing successful architectures (CANN).

2. Unfair experiments

The claims of performance improvement over human-designed CANNs are questionable. For example, the paper states in the abstract: “LLM-generated CANNs achieve accuracy comparable to or greater than manually engineered counterparts, while also exhibiting reliable generalisation to unseen loading scenarios and extrapolation to large deformations.”
However, this performance gain appears to result from _larger network architectures_ chosen by the LLM (e.g., 128–128–64–32 units) compared to the smaller manually designed models (e.g., 12–12 units).  This is the case for the skin data experiment (see ~L426) and for the rubber ones (see ~L372). Increased capacity naturally improves accuracy, but it's just a hyperparameter. Therefore, the comparison as presented is not insightful.

3. Lack of scientific insights

Beyond demonstrating that an LLM can generate working CANN code, the paper offers little analysis or interpretation. There is no exploration of the right way to integrate LLMs for scientific code generation, how prompts influence performance, or which design patterns emerge in the generated code. As a result, the scientific contribution remains minimal and superficial.

Minor comment - the first two figures are not referenced in the text.

**Questions:**

1.  L158 states, “This approach reduces the tensor-to-tensor mapping to a compact scalar regression, enforces thermodynamic consistency, and improves interpretability.”  Could you kindly clarify exactly how CANN enforces thermodynamic consistency or interpretability?
2. _Section 2.3:_ Please explain why CSGA performs worse than CANN. What limitations account for its lower accuracy? Are these models less physics-informed or less constrained, for example? This is relevant as CSGA is one of the primary comparisons.

---

> ### Author Response · Authors · 2025-11-24
> **Comment 1/2**
>
> Thank you for your detailed analysis of our work. We would like to not only answer your questions but also comment on the weaknesses you see:
> - The first and third arguments regarding the weaknesses of our manuscript appear closely related, so we address them together. You note that there is no clear takeaway beyond demonstrating that an LLM can generate working CANNs. However, we consider this result itself to be highly significant. Our paper outlines decades of intensive research on accurate constitutive models and the ongoing challenge of making their generation for new materials more efficient. Demonstrating that LLMs can automatically generate constitutive models is an important contribution, as it enables users with limited background in continuum mechanics to create suitable models quickly and effectively. In addition, on a meta level, constitutive modeling is a particularly demanding domain: models must be built from extremely small datasets (e.g., for rubber, only about 15 data points per loading path), generalize to unseen loading scenarios, and extrapolate beyond the training range while adhering to strict physical constraints such as objectivity and thermodynamic consistency. Our experiments confirm that these requirements can be met. As an application of cutting-edge ML to physical sciences, we believe this constitutes a meaningful contribution.
> Furthermore, we have substantially expanded our analysis of the CANN generation process. The revised manuscript now includes an examination of error modes during generation, a discussion of error handling strategies, statistical evaluation of the refinement steps, and a quantitative assessment of model accuracy across multiple runs. These findings provide insights that extend beyond our specific use case and contribute to understanding how LLMs can be integrated into scientific code generation workflows.
> - You suggest that the performance gains of GenCANNs compared to manually designed CANNs are primarily due to larger network architectures. We would like to clarify that our goal was never to claim superiority over manually engineered CANNs, but rather to demonstrate that users without the expertise to implement a CANN themselves can still obtain effective constitutive models using our framework. The baseline CANNs were intended rather as a reference point to contextualize the performance of GenCANNs. That said, we understand your concern and have addressed it by adding a new set of experiments in which the LLM was explicitly constrained to match the baseline architecture. These results, now detailed in Section A1 with six supporting plots and centrally discussed in Section 5, show that for brain and rubber datasets, constrained and unconstrained GenCANNs perform with similar accuracy. For skin, the unconstrained model benefits from its larger size, yet even the constrained GenCANN still clearly outperforms the baseline. While larger networks can fit training data more closely, they also risk overfitting - especially with small datasets typical in constitutive modeling. Our generalization tests confirm that both constrained and unconstrained GenCANNs extrapolate beyond the training range and generalize to unseen loading states with remarkable accuracy.
> - Thanks for noting that the first two figures were not referenced in the text – we have corrected this now.

---

> > ### Author Response · Authors · 2025-11-24
> > **Comment 2/2**
> >
> > - We thank you for your question on how CANNs enforce thermodynamic consistency and improve interpretability, as this highlights key strengths of our approach. Thermodynamic consistency is ensured through the CANN architecture, which extends beyond a standard feedforward network by incorporating pre- and post-processing steps grounded in continuum mechanics. The second law of thermodynamics requires that the dissipation inequality holds: $D = P : \dot{F} - \dot{\psi}(F) \geq 0$, where $P$ is the first Piola-Kirchhoff stress, $F$ the deformation gradient, and $\psi(F)$ the Helmholtz free energy. For hyperelastic materials, dissipation vanishes $D = 0$, meaning the material is purely elastic and no energy is lost as heat, so the inequality simplifies to: $P : \dot{F} = \dot{\psi}(F)$. From this, the stress follows as: $P = \frac{\partial \psi(F)}{\partial F}$. Rather than approximating the nine stress components directly, the network predicts the scalar free energy $\psi(F)$, and the stress is obtained by differentiating this potential with respect to $F$. Because $P$ is derived from $\psi$ rather than predicted independently, the resulting constitutive model inherently satisfies the entropy inequality and the dissipation condition required by thermodynamic consistency. This step-wise process not only guarantees adherence to physical constraints but also improves interpretability: the neural network’s role is reduced to predicting a single scalar quantity - the strain energy - from strain invariants, rather than a high-dimensional stress tensor. This reduction transforms the problem into a compact, physically meaningful regression task, where the output has clear physical bounds and interpretation, simplifying debugging and enhancing transparency compared to direct tensor-to-tensor mappings.
> > - Your second question concerns why CSGA performs worse than CANNs. A detailed analysis of this issue is provided in the original CSGA paper, which also uses CANNs as a baseline and is cited in our manuscript. In brief, CANNs are highly specialized architectures with refined pre- and post-processing steps that closely mirror continuum mechanics theory. They employ problem-specific approaches, such as using principal stretches or invariants combined with tailored neural network structures, that optimize accuracy but require significant expert knowledge to configure. In contrast, the CSGA approach is much more general: essentially a prompt instructing an LLM to build a constitutive model. This example illustrates that LLM-based constitutive modeling is far from trivial.
> > A further technical factor is that when CSGA incorporates invariants into its constitutive modeling, it introduces a non-convex optimization problem with multiple local minima and collinearity among terms. These characteristics make it substantially harder to achieve a unique global fit compared to the more structured and physics-informed CANN architectures.

---

### Official Review · Reviewer_aqh8 · 2025-11-01

**Soundness:** 2
**Presentation:** 2
**Contribution:** 1
**Rating:** 0
**Confidence:** 4

**Summary:**

This paper presents "GenCANN," a framework where an LLM is prompted to generate and iteratively refine a physics-constrained neural network (CANN) for material modeling. The system is evaluated on three mechanics datasets, where it is shown to match or exceed the performance of "human-designed" baselines.

While the goal of automating scientific modeling is relevant, the paper in its current form is not suitable for publication. Its central claims are invalidated by a combination of critical factors: 1) The problem formulation is a "toy-level" task that is orders of magnitude simpler than established code-generation/agentic (eg, MLE bench) benchmarks, making the LLM's success unsurprising. 2) The experimental methodology is fatally flawed, primarily by failing to account for the stochasticity of LLM outputs and by using confounded, unfair baselines. 3) The work is fundamentally irreproducible as submitted, lacking the necessary artifacts (prompts and conversation logs) to verify its claims.

**Strengths:**

The paper addresses an important and popular application area: the use of LLMs to automate and lower the barrier to entry for complex constitutive modeling tasks.

The proposed framework, which combines LLM-based code generation with a physics-informed constitutive model ML framework (CANN), is clearly presented.

**Weaknesses:**

- The premise of using LLMs as code-generating/agents for scientific tasks is not novel; in fact, the paper's own literature review cites numerous recent examples. The specific task delegated to the LLM is a simple, "fill-in-the-blanks" hyperparameter selection for a small regression model within a predefined, human-authored code skeleton. This is far simpler than rigorous, established benchmarks like SWE-bench (which requires fixing real-world GitHub issues) or MLE-bench (which involves end-to-end ML competitions). Given that LLMs are known to struggle on these complex benchmarks, their success on this paper's highly constrained, simple regression task is entirely expected and provides no new insight.

- The core method involves an LLM in a stochastic 3-hop refinement loop, where the "best-performing version is kept". However, the results in Table 1 are presented as single, deterministic $R^2$ scores. The paper makes no mention of if running this 3-hop-trials multiple repeats, nor does it report any mean, standard deviation, or variance for its results. This is a critical omission. As presented, the results are not scientific findings; they are single, unreplicated cherry-picks or "lucky runs."

- The comparison to a "human-designed CANN" is not clearly stated. The paper states that these baselines are static, pre-existing models taken from prior literature (e.g., "the best CANN reported in the literature (Pierre et al., 2023)" and "the initial CANN publication (Linka et al., 2021)"). This is an "apples-to-oranges" comparison. The GenCANN is a 3-hop search algorithm allowed to optimize for the specific dataset, while the "human" baseline is just a single, static point from previous results.

- The paper's own analysis reveals the confounding variable that likely explains its results: the LLM-generated models are simply massively larger than the baselines. For the Skin dataset, the paper admits the GenCANN's (128-128-64-32 layers) superior accuracy "likely reflects its larger architecture" compared to the baseline's (12-12 layers). The finding that "a much larger network achieves a better R2 score on a regression task" can be trivially found by human too (without nearly no effort) and does not show any advantage of "LLM-auto-design".

- The paper's own data shows that $R^2$ is a flawed and non-robust feedback signal. For the synthetic rubber dataset, Table 1 reports a "perfect" R2=1.00 for both the GenCANN and the baseline. However, Figure 6(c) clearly shows the exact same baseline model failing catastrophically (high relative error) at the x-axis boundaries. The aggregate $R^2$ score completely hides this critical failure. Using this highly-compressed metric to guide the LLM's refinement does not make sense-how did it lead to a better results in Figure 6, while the $R^2$ is also 1 for a poor architecture?

- The work is unverifiable. The authors do not provide the exact prompts or the full conversation logs from the 3-hop refinement process. The "Exemplary GENCANN IMPLEMENTATION" shows only the final product, not the process, and the "Reproducibility Statement" links only to code repositories, not these critical generative artifacts.

**Questions:**

- Did the authors run the 3-hop refinement experiment only once for each dataset? If not, what are the mean and standard deviation of the R2 scores over (e.g.) 10 or 20 independent trials? Without this, the results in Table 1 are meaningless.

- How can the authors justify comparing a 3-hop search algorithm (GenCANN) against a single, static, published baseline (CANN)? A fair baseline would be a human expert given the same 3-hop refinement process, or a standard hyperparameter optimization (e.g., random search) run with a similar budget.

- Given that Figure 6 clearly demonstrates that R2 hides catastrophic model failures at the boundaries, why was this aggregate metric chosen as the sole feedback signal for the LLM, rather than a more robust, physics-aware metric (e.g., max relative error)?

- Will the authors provide the full, unedited prompts and complete conversation logs for all refinement trials to allow for verification of the generative process?

---

> ### Author Response · Authors · 2025-11-24
> **Comment 1/2**
>
> We appreciate your thorough engagement with our work. Here are our responses to your comments, point by point:
>
> - Obviously, our approach goes far beyond simple code generation. A successful constitutive model is not just syntactically correct Python; it must train on very little data (e.g., for rubber only 15 data points per loading path), generalize to unseen loading scenarios, and extrapolate beyond the training range while adhering to physical constraints such as objectivity and thermodynamic consistency. We have confirmed all of this. We do not claim a breakthrough in LLM code generation: our contribution is an effective application to the physical sciences, which is the primary area of research we have selected for our submission. The literature contains many peer reviewed papers that optimize a CANN architecture for a single dataset; we not only outperform these baselines but also provide a generic approach that makes such modeling accessible for new materials to scientists and engineers without deep specialist background.
> - You criticize that we did not report how many times we ran the CANN generation per dataset or provide variance statistics for R². For each dataset we ran the full CANN generation, including three refinement rounds, five independent times and reported the best run in Figures 3–8. We honestly thought we had mentioned this in Section 3, but we missed it and thank you for highlighting this. We have clarified this in Section 3 and added a statistical analysis in Section 4.4 and Figure 9, which summarizes error rates and the stability across runs. In brief, the rubber datasets show no detectable variance, with all runs achieving R² scores of approximately 1; the brain and skin datasets show small variance with the R² scores averaged over loading scenarios remaining well above 0.9. Because the pipeline is partly stochastic, we recommend generating several models and selecting the best one, as we did for our plots.
> - You argue that comparing our CANN generation that includes the described refinement process to a static published CANN was unfair because the LLM is allowed three refinements while the human baseline is not. The opposite is true. As you yourself cited in your comment, for each dataset, we used the strongest human designed baselines available in the literature. For the brain dataset, we identified four peer reviewed studies that explicitly optimized CANNs for this dataset (cited in Section 4.1) and selected the best performing model. For the skin and rubber datasets, one study each conducted comprehensive hyperparameter optimization, and again we used the best reported configurations. We have made this even clearer in the introduction of our baselines in the manuscript. We cannot think of a stronger baseline than multiple expert optimized, peer reviewed CANNs reflecting extensive human search, and we believe this is far stronger than a single engineer allowed only three refinements.
> - You argue that there is no advantage in “LLM-auto-design” when the LLM-generated CANNs outperform manually designed counterparts while containing larger networks. We strongly disagree: the ability to automatically generate constitutive models through LLMs is itself the core contribution, as it enables users with only limited background in continuum mechanics to create suitable models quickly and effectively. That said, we acknowledge the importance of isolating the effect of network size. To address this, we conducted a new set of experiments where the LLM was constrained to match the baseline architecture. These results, now detailed in Section A1 with six supporting plots and centrally discussed in Section 5, show that for brain and rubber datasets, constrained and unconstrained GenCANNs perform with similar accuracy. For skin, the unconstrained model benefits from its larger size, yet even the constrained GenCANN still clearly outperforms the baseline. While larger networks can fit training data more closely, they also risk overfitting - especially with small datasets typical in constitutive modeling. Our generalization tests confirm that both constrained and unconstrained GenCANNs extrapolate beyond the training range and generalize to unseen loading states with remarkable accuracy.

---

> ### Author Response · Authors · 2025-11-24
> **Comment 2/2**
>
> - You state that Figure 6 suggests catastrophic failure at the boundaries for the baseline CANN and question why we chose R² as the feedback signal for the LLM instead of a physics-aware metric such as maximum relative error. First, we want to clarify that the deviations shown in Figure 6(c) are far from catastrophic. In fact, the original peer-reviewed publication of this baseline CANN in the Journal of Computational Physics explicitly notes for this same plot: “the relative error of the CANN is still mostly not higher than around 0.5% and nowhere higher than 3.5%. This demonstrates that CANNs exhibit not only an excellent generalizability but also an excellent ability to extrapolate beyond their training data, likely due to the substantial physical knowledge they incorporate a priori in their architecture” ¹. A prediction error of this magnitude must be considered a remarkable strength of the approach and is nowhere near catastrophic failure. The baseline only appears weak in our manuscript because the LLM-generated CANNs achieve even better generalization and extrapolation, including when constrained to the same network size. Second, regarding the choice of R² as the feedback signal: we are not sure if we understand your question correctly. Simply evaluating the model performance on the training data with the maximum relative error as metric instead of R² as metric would not change anything in the model performance, as that is already close to ideal on the training data and a different metric would not capture inaccuracies beyond the training data either.
> - If we understand your question correctly, it is not really about whether R² or maximum relative error better captures deviations on the training data, as our models perform close to ideal on the training data anyway.  In addition, both metrics would fail similarly to detect significant errors on unseen loading scenarios or outside the training range when not evaluated there. Rather, we interpret your question as asking why we do not include the maximum relative error on the loading paths and stretch regimes that are reserved for testing in the LLM feedback signal. While our synthetic dataset would allow us to use an arbitrary amount of data for training and include many more loading paths and extended stretches in the feedback signal, this does not reflect real-world conditions. In actual laboratory experiments, only a very limited number of tests can be performed, and models must generalize and extrapolate beyond the available data. Our synthetic dataset enables us to verify these capabilities after training, but if we were to include evaluations of generalization and extrapolation in the feedback signal, we would effectively train on data intended for testing, thereby invalidating the experimental design.
> - The prompts have already been included in the GitHub repository under the file name llm_prompts.py. Our work has been verifiable by downloading the referenced datasets and executing the code provided in the repository. In response to your request, we have now also uploaded the complete, unedited conversation logs to the GitHub repository.
>
> 1: Kevin Linka, Markus Hillgartner, Kian P Abdolazizi, Roland C Aydin, Mikhail Itskov, and Christian J Cyron. Constitutive artificial neural networks: A fast and general approach to predictive data-driven constitutive modeling by deep learning. Journal of Computational Physics, 429:110010, 2021. doi: https://doi.org/10.1016/j.jcp.2020.110010.

---

### Author Response · Authors · 2025-12-03
**Discussion summary**

Dear Area Chair,

As the rebuttal phase was aborted and you stepped in, here is a concise summary of the review discussion and the concrete revisions we made. It is unfortunate that the original reviewers cannot evaluate these improvements themselves; we hope this summary helps your assessment.

What reviewers largely agreed on and how we responded:
- Unfair size comparison (GenCANNs larger than baselines): We added a matched capacity study in which the LLM is constrained to exactly the baseline CANN sizes. Findings: on brain and rubber, constrained and unconstrained GenCANNs are indistinguishable and both match/beat the baselines; on skin, the larger (unconstrained) GenCANN fits training paths more tightly, yet the constrained GenCANN still clearly outperforms the baseline; generalization to unseen loading remains strong in both. We added figures and discussion (Appendix A1, Section 5).
- Lack of statistical analysis / accounting for LLM stochasticity: We added a stability study with five independent complete generations per dataset (each with three refinement rounds), reporting mean/variance of R² across loading scenarios, plus failure modes and their rates. Before retries, 31% syntax and 5% training errors occurred; all were resolved by retries/refinement. Rubber shows essentially zero variance across runs; brain and skin show small variance with mean R² well above 0.9. We clarified that plots previously showed the best run and now provide distributional results and a recommendation to run several generations (Section 4.4, Fig. 9).

Reviewer specific notes:
- aqh8 (several concerns that we believe stemmed from misunderstandings, resolved in the revision):
  - Fairness and robustness: We now report multi run statistics and add the matched capacity study, showing results are not driven by lucky runs or larger models; baselines are the strongest, peer reviewed, expert optimized CANNs available for each dataset.
  - Metrics and reproducibility: The “boundary failure” in Fig. 6 is in fact ≤3.5% relative error in the cited baseline and far from catastrophic. We keep R² on training data as the feedback signal to avoid test leakage and evaluate generalization separately. We also released the exact prompts and full, unedited conversation logs for all refinement trials in the repo.
- GdJT:
  - Contribution and capacity: We expanded analysis of the generation process and failure modes (Section 4.4) and added matched capacity results (Appendix A1) showing gains persist beyond size. We corrected missing figure references.
  - Clarifications: We provided a concise explanation of how CANNs enforce thermodynamic consistency (stress from a learned strain energy potential) and why CSGA underperforms (less specialized physics structure).
- JyT9:
  - Process transparency and robustness: We quantified failure modes and refinement benefits; uploaded full conversation logs; updated Fig. 7 to clearly distinguish train/test; explained the CSGA’s unusual mix of weak pure shear fit but reasonable interpolation on mixed biaxial states.
  - Capacity effects: The matched capacity study shows generalization is preserved; the main benefit of larger models appears only on skin training paths.
- cWjb:
  - Scope and “agentic” depth: We clarified that GenCANN automates domain specific decisions (invariants, constraints, fiber handling, boundary conditions) that NAS/AutoML do not cover; we added error handling and stability analyses and discuss extensions beyond hyperelasticity.
  - Capacity concern: Addressed by the matched capacity study; constrained GenCANNs remain competitive or better.

All revisions (new sections, figures), prompts, and full conversation logs are included in the updated manuscript and repository. We appreciate your time and consideration.

---

### Meta-Review · Area_Chair_3PQh · 2025-12-13

**Summary:**

Summary of the reviewer's concerns:
* Baseline comparison. Reviewers highlighted that GenCANNs uses a much larger model, which is an unfair comparison to baselines (aqh8, cWjb, GdJT)
* Human baseline: comparing a learned LLM-based baseline to a human baseline is not fair (reviewer aqh8)
* The choice of R^2 metric that might not be well-behaved near boundaries (reviewer aqh8)
* Evaluating on “toy tasks” while more sophisticated benchmarks exist (reviewer aqh8)
* Limited technical novelty (reviewer cWjb, GdJT)
* Reproducibility and more rigorous evaluation, including reporting the error bars (reviewer jW7S)

**Reviewer Concerns:**

Addressed questions
* Model size: The authors presented the experiments constraining the LLM to match the size of the baseline and reported that constrained LLM still outperformed the baseline.
* Human baseline: The authors argued that they compared to the "strongest human designed baselines available in the literature”
* The authors reported the variance across 5 random seeds.

Unaddressed concerns:
* Toy-level tasks in comparison to other benchmarks like SWE-Bench (reviewer aqh8)
* Limited technical novelty and technical depth. Reviewer cWjb criticized the paper for being a customized API integration rather than a scientific paper.

**Reviewer Scores:**

The key concerns of reviewers about insufficient evaluation and technical novelty remain. The scores likely remain unchanged.

---

### Decision · Program_Chairs · 2026-01-26

Reject